# Reviving Natural Rubber Synthesis via Native/Large Nanodiscs

**DOI:** 10.3390/polym16111468

**Published:** 2024-05-22

**Authors:** Abdul Wakeel Umar, Naveed Ahmad, Ming Xu

**Affiliations:** 1BNU-HKUST Laboratory of Green Innovation, Advanced Institute of Natural Sciences, Beijing Normal University at Zhuhai (BNUZ), Zhuhai 519087, China; 2Joint Center for Single Cell Biology, Shanghai Collaborative Innovation Center of Agri-Seeds, School of Agriculture and Biology, Shanghai Jiao Tong University, Shanghai 200240, China; naveedjlau@gmail.com; 3Guangdong-Hong Kong Joint Laboratory for Carbon Neutrality, Jiangmen Laboratory of Carbon Science and Technology, Jiangmen 529199, China

**Keywords:** rubber polymerase, *Hevea brasiliensis*, native nanodisc, large nanodisc, natural rubber

## Abstract

Natural rubber (NR) is utilized in more than 40,000 products, and the demand for NR is projected to reach $68.5 billion by 2026. The primary commercial source of NR is the latex of *Hevea brasiliensis*. NR is produced by the sequential cis-condensation of isopentenyl diphosphate (IPP) through a complex known as the rubber transferase (RTase) complex. This complex is associated with rubber particles, specialized organelles for NR synthesis. Despite numerous attempts to isolate, characterize, and study the RTase complex, definitive results have not yet been achieved. This review proposes an innovative approach to overcome this longstanding challenge. The suggested method involves isolating the RTase complex without using detergents, instead utilizing the native membrane lipids, referred to as “natural nanodiscs”, and subsequently reconstituting the complex on liposomes. Additionally, we recommend the adaptation of large nanodiscs for the incorporation and reconstitution of the RTase complex, whether it is in vitro transcribed or present within the natural nanodiscs. These techniques show promise as a viable solution to the current obstacles. Based on our experimental experience and insights from published literature, we believe these refined methodologies can significantly enhance our understanding of the RTase complex and its role in in vitro NR synthesis.

## 1. Introduction

Natural rubber (NR), also referred to as cis-1,4-polyisoprene, is synthesized via a sequential cis-condensation mechanism. This intricate process entails the reaction between isopentenyl diphosphate (IPP) and initiators, namely oligomeric allylic pyrophosphate, in the presence of metalloenzymes [1,2,3]. NR synthesis and accumulation occur in a specialized cell organelle called a rubber particle (RP) [4,5,6]. RPs are 88 ± 15 nm average-size spheres of lipid–protein monolayer membrane, based on cryogenic transmission electron microscopy (cryo-TEM), surrounding hydrophobic core polyisoprene chains [2,7]. Based on their size distribution and differential centrifugation, the RPs are divided into small (precipitated with 20,000–50,000× *g*) and large (precipitated with 1000 to 8000× *g*) particles, as shown in Figure 1 [8].

RP-associated proteins interact with the lipid monolayer via a transmembrane domain, lipid–protein covalent bonds, or protein–protein interaction forming functional complexes [2]. Currently, the main enzymes responsible for NR biosynthesis include cis-prenyltransferases (CPTs), cis-prenyltransferase-like (CPTL), rubber elongation factors (REFs), and small rubber particle proteins (SRPPs), all of which are associated with RPs [2,9,10,11,12,13]. The SRPPs and REFs are mainly involved in rubber particle formation and stabilization [14]. As the name indicates, the SRPPs are only found to be associated with small rubber particles (SRPs). The large REF (19.6 kDa) is exclusively found on the large rubber particles (LRPs), while the small REF (14.7 kDa) is equally expressed on both LRPs and SRPs, as shown in Figure 2 [2,15]. It has been demonstrated that SRPs have a higher rubber biosynthesis ratio as compared to LRPs [15], and the down-regulation of SRPPs (exclusively SRP proteins) has affected the integrity of RPs and NR contents [16]. Thus, further research is needed to investigate the possible involvement of the SRPPs and REF 19.6 kDa (solely present on LRPs) in NR synthesis initiation and termination, respectively.

*Hevea brasiliensis* (*H. brasiliensis*), the primary commercial source of NR, has been reported to harbor seven CPTs, one CPTL/Hevea rubber transferase bridging protein (HRBP), eight SRPPs, and nine REFs [2,17,18]. A prior study was conducted to investigate the subcellular localization of various proteins, including SRPP2 [10], REF1 [9], H. brasiliensis rubber transferase 1 (HRT1)/CPT6 [11], and HRBP [12,13]. Their findings implied that HRBP, REF1, and SRPP2 were associated with the endoplasmic reticulum (ER), whereas CPT6 was found to be a cytosolic protein that is recruited to the ER by SRPP2 and the plasma membrane by HRBP when transiently expressed in *Nicotiana benthamiana* [2,12]. Additionally, the co-expression of HRBP-HRT1/CPT7 was observed to be co-localized with Golgi bodies [13]. Protein cargo moving from the ER is processed and modified within the Golgi bodies and sent to the subcellular localization [19]. Thus, hypothetically, we can conclude that both ER and Golgi bodies play a significant role in the NR-producing proteins and enzymes’ posttranslational modification (PTM), transportation, and subcellular localization, which must be considered during the development of any protein synthesis system, especially a cell-free protein synthesis system (CFPSS).

A CFPSS offers numerous advantages, including faster protein synthesis, higher yield, and simplified protein screening [2]. The CFPSS derived from wheat germ extract is known for its productivity and ability to synthesize large amounts of proteins [2,20]. To reconstitute the RTase complex, a prior study employed PCR to amplify the key genes responsible for coding the necessary proteins (HRT1, HRBP, and REF). They cloned them into appropriate vectors for further manipulation. These proteins were synthesized in wheat germ extract containing a CFPSS in the presence of either WRPs or liposomes [4,13]. The proteoliposomes successfully incorporated the RTase complex but did not exhibit any RTase activity in vitro, while the WRPs carrying the reconstituted proteins demonstrated activity, as shown in Figure 3A [13]. The researchers propose that proteoliposomes could be explored further, involving additional candidate proteins and specific lipid molecules, to gain a deeper understanding of rubber particle structure and function [2,13,21,22].

The utilization of CFPSSs supplemented with nanodiscs has greatly facilitated the synthesis of membrane proteins (MPs) and the investigation of their structure and interactions [23,24,25]. Optimization of CFPSSs for different biological systems has been a research subject [26,27]. In an early attempt, a CFPSS supplemented with nanodiscs was used to produce *E. coli* MPs’ functional folds [28]. Studying MPs within supramolecular complexes can provide valuable insights into their structural characteristics, conformational dynamics, and interactions with other proteins and lipids, as well as their topological properties. A highly effective approach for such investigations involves the utilization of a planar bilayer model system, possessing a diameter of approximately 10 nm. This experimental strategy enables the exploration of cross-membrane signaling pathways, thereby enhancing our comprehension of MP functionality [23].

Nanodiscs, which are disc-shaped lipid bilayers stabilized by membrane scaffold proteins (MSPs), have been widely used to investigate protein and membrane lipid structure, function, and interactions [21,23]. The size of nanodiscs is determined by the length of MSPs and the stoichiometry of lipids used in the self-assembly process, resulting in monodispersed particles with similar sizes [29,30,31]. Successful self-assembly of various types of MPs into nanodiscs has been reported, involving the mixing of lipid–detergent micelles and MSPs [23,31,32].

The successful reconstitution of the HRT1-HRBP complex on MSP nanodiscs in a CFPSS has led to the demonstration of CPT activity, marking a significant milestone, as shown in Figure 3B [21]. However, in the case of combining the purified HRT1–nanodisc complex with the HRBP–nanodisc complex, no CPT activity was observed. This observation implies that co-expression is a prerequisite for CPT activity. Kuroiwa et al. (2022) could not achieve RTase activity in either co- or separate expression of HRT1 and HRBP. They realized that other important proteins, such as REF and SRPPs, must be required for in vitro RTase activity. Hence, the researchers co-expressed them on nanodiscs with the HRT1-HRBP complex. However, due to the low affinity of REF and SRPPs to the nanodiscs’ membranes or flat lipid bilayer structures, they were lost during the purification step [21].

In summary, CFPSSs have provided significant advantages for studying the structure and function of membrane-associated, potentially including RP-associated, proteins. However, our understanding of the techniques for incorporating RP-related proteins needs improvement. We must consider various factors to achieve in vitro NR synthesis via the reconstitution of the complete RTase complex and associated proteins on nanodiscs in a CFPSS. One of the known factors to be addressed is the low affinity of SRPPs and REFs to nanodisc membranes or lipid bilayer structures. Similarly, the conditions optimization for co-expression and reconstitutions of known proteins potentially involved in NR synthesis, exploring alternative lipid compositions, and effective reconstitution process are also critical obstacles to be handled.

In this regard, Umar et al. provided a comprehensive review elucidating the intricate mechanisms underlying natural rubber biosynthesis [2]. Their work encompasses a detailed model illustrating the organization of phospholipids and glycolipids around polyisoprene chains, alongside the identification and characterization of key proteins, including CPTs/HRTs, CPTLs/HRBP, REFs, SRPPs, and novel proteins like Rubber Unusual Proteins (RULP, RUSPI, and RUSPF). Through a series of experimental approaches involving recombinant synthesis in *E. coli*, yeast, and *Arabidopsis thaliana* cells, as well as in vivo reconstitution strategies, the functional roles of these proteins have been studied. Furthermore, insights gained from CFPSSs have shed light on the specific activities of protein complexes on the WRPs and nanodiscs, delineating their roles in synthesizing natural rubber or prenyl chains. Umar et al.’s review provides a comprehensive overview of the molecular machinery driving natural rubber production, advancing our understanding of this industrially significant process.

Considering the comprehensive work presented by Umar et al. [2], it becomes evident that novel approaches are imperative for addressing the challenges surrounding the RTase complex and rubber synthesis machinery. These approaches may involve innovative strategies such as advanced protein engineering techniques, targeted mutagenesis studies, or the exploration of alternative biosynthetic pathways. By harnessing the insights garnered from this extensive review, researchers can embark on novel avenues aimed at enhancing the efficiency and productivity of natural rubber production, thereby contributing to the sustainable advancement of this vital industry. In addition, exploring alternative in vitro CFPS approaches, such as employing nanodiscs, native nanodiscs, and large nanodiscs using detergent-free approaches, could prove to be effective. These techniques offer promising avenues for reconstituting protein complexes in a controlled environment, potentially facilitating a deeper understanding of their biochemical properties and interactions. By harnessing the advantages of these innovative methodologies, researchers can overcome existing challenges in studying the RTase complex and rubber synthesis machinery, thereby advancing our knowledge and paving the way for enhanced natural rubber production. 

**Figure 3 polymers-16-01468-f003:**
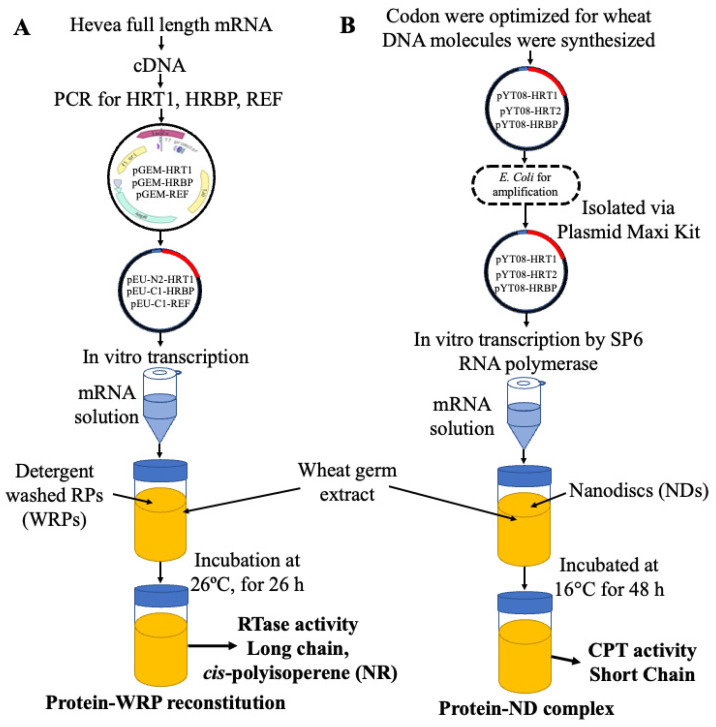
The reconstitution of candidate RTase complexes was investigated using two distinct approaches: (**A**) employing detergent-washed rubber particles [13] and (**B**) utilizing nanodiscs [21,32]. In method (**A**), the HRT1, HRBP, and REF genes were cloned into an expression vector, followed by in vitro transcription and subsequent reconstitution of the HRT1-HRBP-REF complex on detergent-washed rubber particles (WRPs). This complex exhibited substantial RTase activity. In method (**B**), the cloning process of HRT1, HRBP, and REF genes into an expression vector was accompanied by amplification in *E. coli*, isolation of the amplified plasmids, in vitro transcription, and reconstitution of the HRT1, HRT2, and HRBP complexes on nanodiscs. Despite displaying significant PTase activity, this complex failed to showcase RTase activity in the synthesis of long polyisoprene chains or natural rubber.

## 2. CRISPR/Cas9 Mutagenesis: CPT Alone Does Not Solely Determine NR Length

Molecular genetic tools are a prerequisite for the biotech-enabled synthesis of NR, which is highly problematic for well-known NR-producing plants such as rubber trees, guayule, and dandelion [33]. *Lactuca sativa* (lettuce), an easily cultivable, self-pollinating diploid with a 4–5 month life cycle, produces NR in laticifer cells, in a similar way to rubber trees, with an average Mw of >1000 KDa [34,35]. Its transformation amenability and genome sequence accessibility make lettuce highly suitable for CRISPR/Cas9 genome editing to investigate the underlying molecular mechanisms involved in the biosynthesis of NR [33,36]. CPT, a small gene family translating into NR and other cis-polyisoprene biosynthesis enzymes, is further divided into prokaryotic and eukaryotic subgroups [2,33,37,38]. Prokaryotic CPTs, mainly present in bacteria and plant chloroplasts, are soluble and form homodimers. They biosynthesize polyisoprenyl diphosphates, which are converted into undecaprenols in bacteria and polyisoprenols in plant plastids [39,40]. In bacteria, undecaprenol serves as a glycosyl carrier during peptidoglycan cell wall biosynthesis [41,42]. While in plant plastids, polyisoprenols participate in photosynthesis and the fluidity of thylakoid membranes [33,43].

Eukaryotic CPTs are membrane proteins (MPs) that form complexes with CPT-binding protein (CBP) on the ER’s cytosolic side, as discussed earlier [2,4,44]. CBP, a highly conserved eukaryotic protein with a weak sequence homology to CPT, recruits and facilitates active CPT recruitment to the ER and other membranes [12,13,33]. In all eukaryotes, the CPT-CBP complex catalyzes dehydrodolichyl diphosphate (DHDD) biosynthesis, a precursor of dolichol monophosphate (DMP). DMP serves as a sugar carrier during eukaryotic proteins glycosylation; thus, any defect in the CPT-CBP complex, or individually in CPT or CBP that fails glycosylation, is lethal in all eukaryotes [45,46]. The Arabidopsis lew1 (CBP) mutant exhibits impaired membrane integrity but still survives, suggesting that dolichols may have additional functions beyond protein glycosylation in plants [47]. AtCPT1, a homomeric CPT producing long-chain polyisoprenoids, does not need LEW1 for its activity, nor form any complex with the homologues of NgBR/NUS1 [48].

The CPT and CBP direct interaction is demonstrated via yeast two-hybrid [13] or pull-down assays [49]. In *Taraxacum brevicorniculatum* (dandelion) and lettuce, the silencing of laticifer-specific CPT and CBP genes (TbCPT1/2/3 or TbCBP and LsCBP2, respectively) via RNAi silencing resulted in a considerable reduction in NR production [38,50,51]. The isoforms of CPT and CBP have evolved for NR synthesis in certain plant species, such as rubber trees, guayule, lettuce, and Russian dandelion [33,37,51]. The CBP second isoform (LsCBP2) is found in lettuce, PaCBP in guayule, and HbCBP in rubber tree laticifer cells [13,51,52]. While in some plants, CPT has diverged to be NR biosynthesis specific, the divergence, gene duplication, and function of CBP have not been observed in NR-producing plants [33,51].

In lettuce, three CPTs (LsCPT1-3) and two CBPs/CPT-like (CPTL1-2) have been studied previously [51]. Plastid-specific prokaryotic LsCPT2’s function in lettuce is unknown; however, with a low and ubiquitous expression of the eukaryotic LsCPT1 in different tissues, its recombinant proteins have shown efficient DHDD biosynthesis [4,33]. The eukaryotic LsCPT3-LsCBP2 complex is expressed exclusively in latex-producing laticifer cells; however, the in vitro microsomal reconstitution of it could not synthesize NR [33]. Lettuce uniquely encodes a single copy of only three distinctive CPTs, which makes it ideal for CRIPR/Cas9-induced targeted mutagenesis-based function identification [33,53]. As mentioned in the introduction section, other than CPT and CBP, other RP-associated proteins (REF and SRPP) are reportedly involved in NR biosynthesis. However, SRPP-silenced Russian dandelion (*T. kok-saghyz* and *T. brevicorniculatum*) showed only a 40–50% decrease in NR synthesis, whereas lettuce showed no reduction [16,51,54]. In contrast, CPT- or CBP-silenced lettuce and Russian dandelion plants showed nearly no NR biosynthesis [38,50,51]; hence, the previously reported catalytic activities of REF and SRPP have been disproved [9,10].

CRISPR/Cas9-enabled knockout of LsCPT3 resulted in the first NR-deficient mutant (lscpt3), providing strong evidence for its exclusive role in NR biosynthesis [33]. This NR-deficient mutant was used to investigate the catalytic properties of the CPTs in different plant species and their role in NR polymer length. The heterologous expression of guayule and goldenrod CPTs in lscpt3-3 background and the resulting transgenic lettuce expressing LsCPT3, PaCPT3, and ScCPT3 displayed a reduced quantity of NR as compared to wild-type lettuce. As mentioned earlier, eukaryotic CPTs require CBP as a binding partner for dolichol and NR biosynthesis [13,44,51]. The stoichiometric imbalances of transgenic CPT with native CBP, transgenic CPT transcript/protein stability, transcription/translation efficiency, and their positional effects in transgenic lettuce can be the reason for NR reduction [33]. Interestingly, transgenic PaCPT3 and ScCPT3 in the lscpt3-3 background produced longer NR polymers (with more Mw) as compared to wild-type guayule and goldenrod [33]. While transgenic LsCTP3 in the lscpt3-3 background (used as a control) biosynthesized 11–16% shorter NR polymer as compared to wild-type lettuce [33]. Therefore, the Mw and length of the polymer are not only controlled by CTP’s innate catalytic activity but also the host cellular context. Other endogenous elements, such as the availability of extender (IPP) and initiator (FPP) molecules, and suitable organelles with appropriate lipid and protein structures, such as Hevea RP, could be the reason for the variance in NR length in different species [2,4,13,21,32,55]. Exogenous stimuli, such as light/dark period, moisture, water, nutrient availability, temperature, etc., also greatly influence the production of secondary metabolites [56,57,58], especially NR and the length of the polymers in different species [4,59,60,61].

## 3. Cell-Free System for Membrane Proteins

The preparation of MPs, which are generally lipid-embedded, requires laborious and complex methods as compared to water-soluble proteins [32]. Recently, researchers adopted a flexible and alternative open CFPSS for the isolation, preparation, and functional analysis of MPs [26,62]. The inclusion of surfactants or nanodiscs (artificial lipid membrane structures) to the CFPS reaction mixture has been useful for MP investigations, including G-protein-coupled receptors (GPCRs), ion channels, and enzymes [21,27,63,64,65]. A prior work reconstituted NR synthesis and cis-prenyltransferase activity by manipulating candidate genes, HRT1 and HRBP, using a nanodisc-enriched CFPSS [21]. REF/SRPP provide stability or support the formation of RPs and hydrophobic lipid monolayer particles (LMPs) in plant cells [4,66], which are important in cis-prenyltransferase activity [2,32]. A LMP-based CFPSS, as a novel technology for protein expression on LM structures, was developed by testing various lipid compositions compatible with target proteins [32]. The choice of squalene or triacylglycerol (TAG) for the inner hydrophobic core was based on its chemical uniformity and potential resistance to metabolic enzymes in the CFPS reaction mixture [32].

Nanodisc- or LMP-enriched wheat germ CFPSSs have been used to test the in vitro co-translation of REF and SRPP [32]. LMP-binding proteins, such as guayule homolog of SRPP (GHS) [67], Chrysanthemum morifolium carotenoid cleavage dioxygenase 4a (CmCCD4a) [68], and Homo sapiens perilipin 3 (PLIN3) [69], were also tested for validation of co-translation of proteins on LMPs. The *H. brasiliensis* proteins (HRT1, HRBP, REF, and SRPP) as well as homologs, such as PaCPT3, PaCBP, GHS, CmCCD4a, and PLIN3, were synthesized in the presence of LMPs in the CFTS system. The LMP fraction was isolated by centrifugation and nanodiscs by immobilized metal affinity chromatography, and protein reconstitution was confirmed by SDS-PAGE [21,32]. It was further confirmed by radiolabeled [^14^C]L (Leucine or Leu) in the translation mixture [32,70].

The turbidity (OD_600_) of the negative control (LMPs without mRNA template), HRT1-LMPs, HRBP-LMPs, PaCPT3-LMPs, PaCBP-LMPs, and CmCCD4a-LMPs decreased in the suspension and fractionation. On the other hand, the OD_600_ value for the REF-LMPs, SRPP-LMPs, GHS-LMPs, and PLIN3-LMPs suspensions was significantly higher. These results indicate the collapse of LMPs in the CFPS reaction mixture, whereas the LMPs’ structure was significantly stable in the presence of REF, SRPP, GHS, or PLIN3 [32]. Furthermore, HRT1, HRBP, PaCPT3, PaCBP, GHS, and CmCCD4a showed affinity to the nanodiscs, while REF, SRPP, and PLIN3 showed affinity to LMPs. The structural models of REF, SRPP, and GHS created by AlphaFold2 showed that amphiphilic helices enriched with hydrophobic residues in specific positions are likely to interact with LMs, similar to lipid droplet-related proteins [32,71,72]. The REF and SRPP exhibited an affinity towards LMPs; however, they did not show any preference for nanodiscs, much like the PLIN3 [32,73]. Hence, these results provide significant evidence that LMPs effectively mimic lipid monolayer properties in vivo. The fresh LMPs were found to have a 225.5 ± 46.0 nm diameter, while azolectin liposomes had a diameter of 49.4 ± 12.5. Although the form of the LMPs in the CFPS reaction mixture in the absence of mRNA construct was compromised, they were homogeneously well dispersed in the presence of REF and GHS, which successfully prevented LMP aggregation [32]. Consistently, the additional expression of REF in the CFPSS inhibited the aggregation of WRPs [13]. The LMPs associated with REF are approximately the size of LRPs and bigger than SRPs. As REF is ubiquitously present on SRPs and LRPs and SRPP is only present on SRPs, their particle size is hard to replicate [8,21].

The GHS-LMPs structure was found to be more stable and used for the reconstitution of the guayule prenyltransferase complex [32]. The co-expression of PaCPT3 and PaCBP exhibited a significant increase in prenyltransferase activity. In the absence of GHS, the reconstitution of the PaCPT3/PaCBP complex also showed prenyltransferase activity. However, it was difficult to obtain a homogeneous suspension of LMPs [32]. The presence of GHS did not make any difference in the prenyltransferase activity when reconstituted on the nanodiscs [21].

## 4. Detergent-Free Native-like Membrane Proteins’ Reconstitution in Liposome

Protein isolation and the excision of membrane proteins from their native membrane environment lead to protein destabilization and alter their functional properties [74,75]. Various methods, such as employing amphipathic polymers known as “amphipols” instead of detergents, are being explored to isolate membrane proteins in their native-like state [76]. Lipid nanodiscs with membrane scaffolding proteins are used for functional and structural investigations [77]. Another commonly practiced approach that enables detailed functional and structural studies under restored native-like conditions involves reconstituting purified membrane proteins in lipid vesicles [78,79].

All these methods depend on detergents for membrane protein isolation. The selection of a suitable detergent can sometimes yield pure, intact, and active protein complexes [79]. However, developing a universally applicable technique to extract and purify native-like membrane protein or protein complex/es and transfer them to membranes with defined lipid compositions is challenging [79]. A potential solution to this challenge entails utilizing an amphipathic styrene-maleic acid (SMA) copolymer, which allows for the extraction and isolation of the desired membrane protein along with a native lipid disc, eliminating the necessity for detergent [80]. Native nanodiscs typically have a diameter ranging from 10 to 24 nm, while the SMA copolymer has a thickness of approximately 1 nm [81]. Recently, this approach has been successfully used to isolate various membrane-associated proteins, such as proton pumps, photosynthetic systems, and ion channels [79,82,83]. Consistently, the transfer of an ion channel, isolated in native nanodiscs, to a planar membrane resulted in a transmembrane ion conductance [84]. Smirnova et al. (2016) extracted and purified *Saccharomyces cerevisiae* mitochondrial cytochrome c oxidase (CytcO) in its native lipid environment using the SMA co-polymer and affinity chromatography, eliminating the use of detergents. The CytcO complex has been successfully incorporated into liposomes with diameters of ~30 nm or ~110 nm, with the reconstituted complex actively generating the proton electrochemical gradient [79,85].

We propose an innovative, detergent-free technique for reconstituting rubber polymerase within lipid vesicles (liposomes). Drawing inspiration from the above-mentioned recent membrane protein research advancements, this approach utilizes the amphipathic SMA copolymer to extract and purify the polymerase complex while preserving its native lipid environment. Through affinity chromatography and SMA extraction, the purified polymerase complex could be incorporated into defined lipid compositions within liposomes, offering a controlled environment that mimics its natural surroundings. Building on successful methods used for membrane-associated proteins, this technique holds promise for a more accurate study of rubber polymerase’s structure and function. This approach provides valuable insights into rubber synthesis mechanisms and potentially contributes to enhancing natural rubber production, as shown in Figure 4A. At the in vitro scale, the expression and reconstitution of the RTase complex on liposomes ensure the functional integrity of the RTase complex within the liposomal environment. However, unfortunately, it did not exhibit any CPT activity, as shown in Figure 4B [13].

## 5. Large-Sized DNA-Corralled Nanodiscs for Investigating Membrane Proteins

The use of MSP-based nanodisc systems for structural studies, especially for large membrane protein complex/es/arrays or in vitro reconstitution for biological processes, has been limited. Recently, Zhao et al. engineered covalently circularized nanodiscs (cNDs), where amphipathic helical polypeptides (AHP) surround a phospholipid bilayer [86]. The AHP is derived from Apolipoprotein A1 and can be circularized by sortase A [87]. This technique resulted in engineering covalently closed homogenous nanodiscs with ∼8, 11, 15, and 50 nm diameters. These cNDs are highly useful in investigating small- and medium-sized membrane proteins and the interaction of membrane-bound receptors with virus particles [87,88,89,90]. Using a similar approach for synthesizing larger nanodiscs was difficult due to their susceptibility to aggregation and the requirement for large scaffold proteins, which are hard to express and purify in the *E. coli* system [86]. External DNA-origami barrels were applied as scaffolding corrals to synthesize larger nanodiscs [86].

Scaffolded DNA origami self-assembles customized shapes from nanostructures by the folding of a long single-stranded DNA (ssDNA) molecule with the help of a scaffold strand, a shorter complementary DNA strand, that acts as a guide or template [86,91,92]. DNA origami arranges biomolecules with required compositions and stoichiometries for biofunctional studies [93,94]. The precise morphology and dimensions of nanomaterials, encompassing both hard inorganic and soft biological materials, can be constrained by the mechanical stiffness of self-assembled DNA nanostructures during casting growth [95]. A previous study reconstituted nanodiscs with approximately 11 nm diameters, circumscribed by a pair of non-circularized oligonucleotide-functionalized scaffold proteins, into a single large nanodisc [86]. The diameter of a large nanodisc depends on the size of the barrels. In their investigations, two barrels of distinct sizes, 90 and 60 nm, were employed, leading to the formation of DNA-corralled nanodiscs (DCNDs) with approximate diameters of 70 nm and 45 nm, respectively.

Yamashita et al. [13] reconstituted a candidate complex (HRT1-HRBP-REF) to detergent-washed RPs and achieved in vitro NR synthesis. Hence, it could be incorporated into nanodiscs to confirm this complex can solely perform rubber polymerase activity. Similarly, Kuroiwa et al. [21] reconstituted HRT1-HRBP on MSP–lipid nanodiscs in a CFPSS, and they achieved significant CPT activity. However, they could not achieve RTase activity and tried to co-express REF and SRPPs with HRT1-HRBP on the nanodiscs. Due to their low affinity to nanodisc membranes or flat lipid bilayers, they were lost during the purification step. As such, incorporating the Yamashita complex (HRT1-HRBP-REF) with SRPPs into large nanodiscs may solve this problem. The proposed process for the incorporation of the RTase complex into the large nanodiscs is shown in Figure 5.

## 6. Conclusions

In summary, this review underscores the critical importance of understanding the rubber transferase (RTase) complex, which plays a central role in the synthesis of natural rubber (NR). Despite numerous efforts, isolating and characterizing this complex remains a significant challenge. The proposed approach, utilizing detergent-free methodologies with natural nanodiscs, offers a promising solution. By integrating the RTase complex into liposomes or large nanodiscs, the aim is to bridge the gap between in vitro studies and cellular environments, facilitating deeper insights into NR biosynthesis. Drawing upon a comprehensive synthesis of literature and hands-on experimental expertise, the advocacy lies in these refined methodologies as catalysts for paradigm-shifting advancements in RTase research. Through these methods, significant enhancements in the understanding of NR synthesis mechanisms are anticipated, paving the way for more efficient in vitro production methods. Thus, these efforts are dedicated to addressing the complexities involved in RTase complex research, with the ultimate goal of advancing NR synthesis and meeting the increasing global demand.

## Figures and Tables

**Figure 1 polymers-16-01468-f001:**
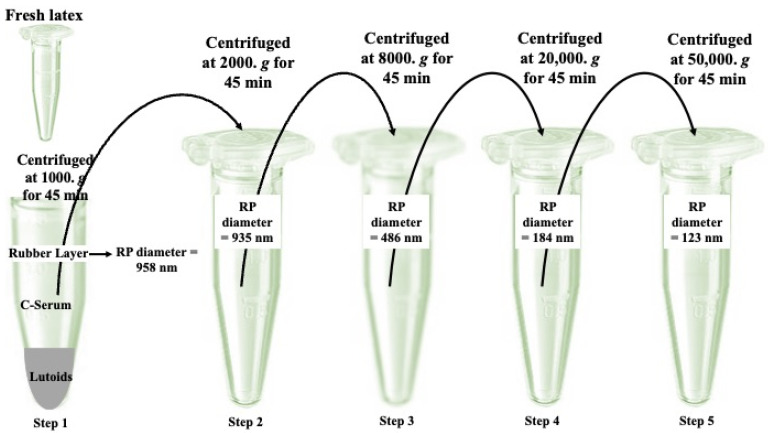
The latex components are separated into rubber particles of varying sizes using a modified version of the method described by Yamashita et al. [8]. The freshly collected latex from the rubber tree undergoes a precise procedure within a controlled buffer environment. Through a series of carefully executed centrifugation steps, distinct fractions are isolated, and rubber particles (RPs) of different sizes are obtained by applying varying centrifugal forces. The stepwise centrifugation technique is conducted as follows: Step 1: The fresh latex is subjected to centrifugation at 1000× *g* for 45 minutes (min) to eliminate excessively large RPs and aggregated rubber. Step 2: The C-serum fraction is then transferred to a new tube and centrifuged at 2000× *g* for 45 min. The resulting RPs, located in the upper layer known as the rubber layer, possess an average diameter of approximately 935 nm. Step 3: The C-serum fraction obtained from step 2 is centrifuged at 8000× *g* for 45 min, yielding isolated RPs ranging in size from 486 nm. Step 4: Subsequently, the C-serum fraction from step 3 is centrifuged at 20,000× *g* for 45 min, resulting in isolated RPs with a size of 184 nm. Step 5: Finally, the C-serum fraction obtained from step 4 is centrifuged at 50,000× *g* for 45 min, leading to the isolation of RPs with a size of 123 nm.

**Figure 2 polymers-16-01468-f002:**
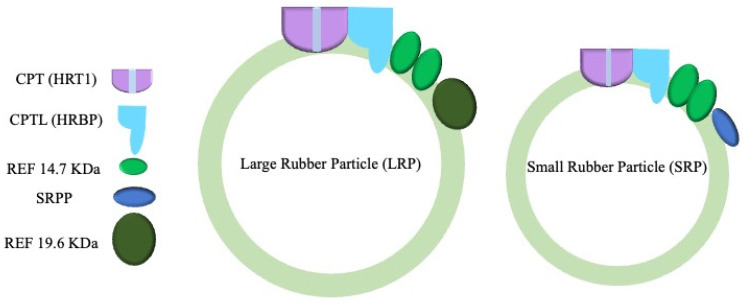
The composition of the biosynthetic apparatus and the disparities in candidate membrane proteins are investigated in large and small rubber particles [2,4,8]. These proteins encompass cis-prenyltransferases (CPTs), the putative Hevea rubber transferase 1 (HRT1), Hevea rubber transferase 1-REF bridging protein (HRBP)/cis-prenyltransferase-like (CPTL), and small rubber particle protein (SRPP). CPT, CPTL, and REF 14.7 KDa are detected in both LRPs and SRPs, whereas REF 19.6 KDa solely occurs in LRPs and SRPPs within SRPs.

**Figure 4 polymers-16-01468-f004:**
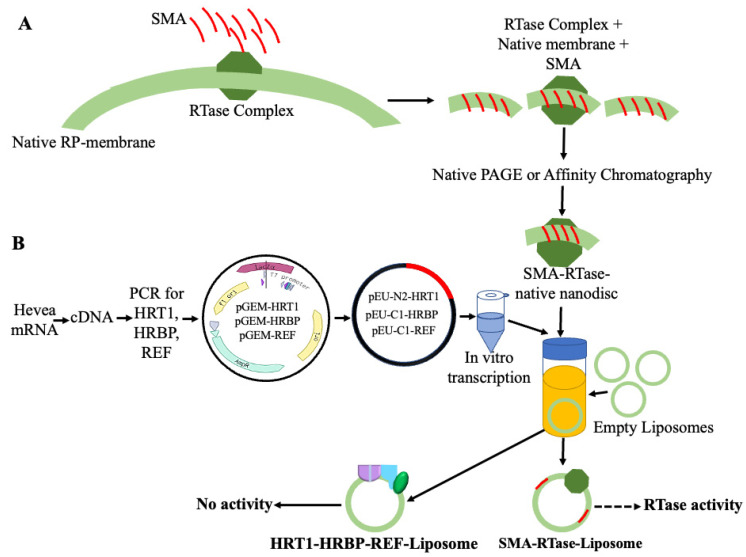
Schematic representation of the proposed process for creating detergent-free natural nanodiscs for reconstituting the RTase complex on liposomes based on the method previously reported [79,85]. (**A**) Isolation of the RTase complex with styrene-maleic acid (SMA) and native membrane: To begin, the RTase complex will be isolated from its natural membrane environment. We will use SMA, which is known for its ability to solubilize membrane proteins, to encapsulate the RTase complex and form SMA–lipid nanodiscs. These nanodiscs typically contain other associated membrane components and serve as the initial material for subsequent reconstitution processes. Next, the isolated RTase complex encapsulated in SMA–lipid nanodiscs will be integrated into liposomes, which are synthetic vesicles made up of lipid bilayers. This integration process involves incubating the nanodiscs with liposomes under specific conditions to promote their fusion or insertion into the liposomal membrane. As a result, the RTase complex becomes incorporated into the liposomal membrane, establishing a foundation for further experimental manipulations. (**B**) In vitro transcription and reconstitution of the RTase complex on liposomes: In this step, the target mRNA will be converted into cDNA. The cDNA is then amplified using PCR, followed by the preparation of transformation and transcriptional constructs. After performing in vitro transcription, the mixture is added to the CFPSS containing liposomes for translation and reconstitution on liposomes. This reconstitution process ensures the functional integrity of the RTase complex within the liposomal environment. However, unfortunately, it did not exhibit any CPT activity.

**Figure 5 polymers-16-01468-f005:**
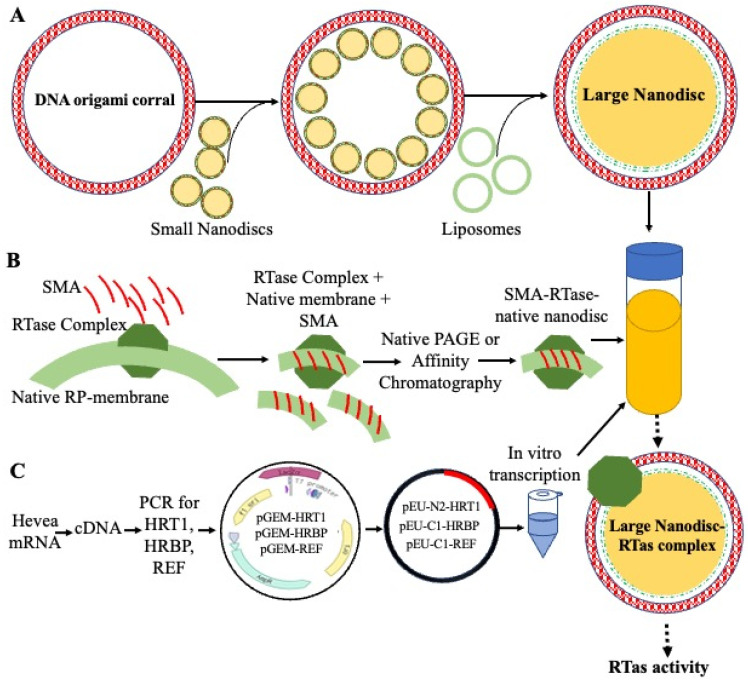
The proposed method for detergent-free RTase complex reconstitution on large nanodiscs. (**A**) Small nanodiscs are initially prepared and functionalized with oligos for specific binding. A DNA origami barrel is then assembled, featuring binding sites tailored for the small nanodiscs. These small nanodiscs selectively bind to their designated sites on the DNA origami structure. Subsequently, detergents and lipids are introduced, facilitating the formation of larger nanodiscs around the bound ones. Through a dialysis process, detergents are gradually removed, leaving behind stable and homogeneous large nanodiscs encapsulated within the DNA origami. Finally, the resulting nanodiscs are subjected to characterization to assess their size, stability, and homogeneity. This comprehensive process yields large nanodiscs suitable for various applications, including membrane protein studies [86,87,96,97]. (**B**) Isolation of the RTase complex with styrene-maleic acid (SMA) and native membrane modified version of [79,85]. To begin, the RTase complex will be isolated from its natural membrane environment. We will use SMA, which is known for its ability to solubilize membrane proteins, to encapsulate the RTase complex and form SMA–lipid nanodiscs. These nanodiscs typically contain other associated membrane components and serve as the initial material for subsequent reconstitution processes. Next, the isolated RTase complex encapsulated in SMA–lipid nanodiscs will be integrated into the large nanodiscs prepared in (**A**) via the CFPSS. (**C**) In vitro transcription and reconstitution of the RTase complex on large nanodiscs via a modified version of [13]. In this step, the target mRNA will be converted into cDNA. The cDNA is then amplified using PCR, followed by the preparation of transformation and transcriptional constructs. After performing in vitro transcription, the mixture is added to the CFPSS containing large nanodiscs for translation and reconstitution.

## Data Availability

The data presented in this study are available on request from the corresponding author.

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
