# Peer review of "Reviving Natural Rubber Synthesis via Native/Large Nanodiscs"

_polymers, 2024, doi:10.3390/polym16111468_

Round 1

Reviewer 1 Report

Comments and Suggestions for Authors

After thoroughly examining the submitted manuscript, I find that it does not fully meet the requirements expected of a review article. Below, I detail the primary concerns and recommendations that must be addressed to enhance the manuscript’s quality and alignment with the journal’s standards.

  1. The manuscript falls short of fulfilling the fundamental purpose of a review article. There is a noticeable lack of comprehensive analysis of existing scientific reports. It fails to identify key gaps and ongoing challenges within the relevant field. I recommend a detailed integration of recent and crucial studies to provide an inclusive overview of the topic.
  2. The paper’s title does not accurately reflect the manuscript's main objectives and content. 
  3. The abstract lacks clarity, which is crucial for summarizing the essence of the review. It should be rewritten to clearly state the main focus of the review.
  4. The introduction does not present a clear progression of ideas, and some content, although relevant, is difficult to follow. Additionally, there is ambiguity regarding Figure 1, which is referenced but not clearly tied to the source cited (Yamashita et al., 2018). Please clarify this citation or correct the figure’s attribution.
  5. Figures 3-5 are included in the document without adequate description or reference within the main text. Each figure should be clearly described and discussed to justify its inclusion and to illuminate its relevance to the text.
  6. The manuscript includes unpublished results without proper justification of the inclusion and exclusion criteria used. For a review article, it is essential to rely on peer-reviewed and published data to maintain credibility and reliability.
  7. The conclusions do not adequately summarize the main insights or contributions of the review. They must relate directly to the objectives outlined in the introduction and reflect upon the analysis presented throughout the article.

While the manuscript contains valuable information pertinent to the field, these substantial issues must be addressed before it can be considered for publication. 

Comments on the Quality of English Language

The manuscript requires significant improvements in language use, from the abstract to the conclusion. The current state of the text makes it difficult to understand the main ideas clearly. A thorough grammatical revision by a native English speaker or a professional language editing service is strongly advised.

Author Response

Response to Reviewer 1

We sincerely appreciate your thorough review and insightful feedback throughout this process. Your valuable input has undoubtedly strengthened our manuscript, and we are grateful for the opportunity to improve our work with your guidance. Here are the responses to each of your comments one by one:

Comments and Suggestions for Authors

After thoroughly examining the submitted manuscript, I find that it does not fully meet the requirements expected of a review article. Below, I detail the primary concerns and recommendations that must be addressed to enhance the manuscript’s quality and alignment with the journal’s standards.

  1. The manuscript falls short of fulfilling the fundamental purpose of a review article. There is a noticeable lack of comprehensive analysis of existing scientific reports. It fails to identify key gaps and ongoing challenges within the relevant field. I recommend a detailed integration of recent and crucial studies to provide an inclusive overview of the topic.

Response:

Thank you for your helpful feedback. We value your ideas and suggestions, which have been incredibly useful in improving our manuscript. We're happy to say that we've addressed the issues you raised by including a detailed summary of our recent review article on the Rubber polymerase complex. This article covers research spanning many decades in the field.

Our team, with more than twenty years of experience in this area, carefully looked through existing research to give a comprehensive understanding of the topic. We've also introduced new approaches to explain the complexities of the RTase complex better. It's important to mention that our revised manuscript doesn't include any new, unpublished work; instead, it brings together ideas from previously published literature.

We think these changes have improved the clarity and importance of our manuscript, making it easier to understand and more helpful for future research. Again, we're very grateful for your helpful feedback, and we're confident that the revised manuscript will make a meaningful contribution to advancing knowledge in this field.

  1. The paper’s title does not accurately reflect the manuscript's main objectives and content. 

Response:

The title of the manuscript captures its main goal: to find new ways to study how natural rubber (NR) is made. It hints at a fresh approach by talking about using both small and big nanodiscs. "Reviving Natural Rubber Synthesis" suggests a renewed interest in this area of research, while "Native/Large Nanodiscs" points to the special techniques being used.

In summary, the title invites readers to explore the manuscript and learn about the exciting strategies proposed to improve our knowledge of NR synthesis.

  1. The abstract lacks clarity, which is crucial for summarizing the essence of the review. It should be rewritten to clearly state the main focus of the review.

Response:

We have revised the abstract to enhance its clarity and readability, ensuring that it effectively summarizes the main focus of the review.

  1. The introduction does not present a clear progression of ideas, and some content, although relevant, is difficult to follow. Additionally, there is ambiguity regarding Figure 1, which is referenced but not tied to the source cited (Yamashita et al., 2018). Please clarify this citation or correct the figure’s attribution.

Response:

We have considered your comments and made revisions accordingly. In response to your concern about the clarity of the citation and attribution of Figure 1, we have included the references in the legends of the figure to provide additional context and clarity. Additionally, we have made it explicit in the text that we are referencing Yamashita et al. (2018) in the context of rubber particle size distribution, particularly relevant to our discussion on the centrifugation-based separation of different-size particles. We appreciate your attention to detail and aim to ensure that our references and attributions are clear and accurate for all readers.

  1. Figures 3-5 are included in the document without adequate description or reference within the main text. Each figure should be clearly described and discussed to justify its inclusion and to illuminate its relevance to the text.

Response:

We ensured that each figure is adequately cited within the main text of the revised manuscript. This ensures clarity and relevance, helping readers better understand the significance of each figure. Furthermore, we have addressed your concern by completely rewriting the figure legends, providing further elaboration, and including relevant references within the legends themselves. Your input has been instrumental in enhancing the clarity and coherence of our work.

  1. The manuscript includes unpublished results without proper justification of the inclusion and exclusion criteria used. For a review article, it is essential to rely on peer-reviewed and published data to maintain credibility and reliability.

Response:

Regarding your concern about unpublished results, I'd like to clarify that our manuscript does not include any unpublished data. Instead, we have extensively utilized previous literature to discuss techniques and their success in studying similar complexes. Specifically, we highlighted that while the structure of the RTase complex bears similarities to other complexes, its unique function and the intricate relationship between its components remain incompletely understood. By drawing parallels with previously successful methodologies in related fields, we proposed the application of similar techniques to address the challenges associated with studying the RTase complex. We aim to leverage established methodologies to advance our understanding of the RTase complex and its role in natural rubber synthesis.

  1. The conclusions do not adequately summarize the main insights or contributions of the review. They must relate directly to the objectives outlined in the introduction and reflect upon the analysis presented throughout the article.

While the manuscript contains valuable information pertinent to the field, these substantial issues must be addressed before it can be considered for publication. 

Response:

We appreciate your insightful comments on our review manuscript. Taking your feedback into account, we have carefully revised the conclusion section to better address your concerns. Specifically, we have reworked it to effectively summarize the main insights and contributions of the review, while directly relating to the objectives outlined in the introduction and reflecting upon the analysis presented throughout the article.

These revisions have significantly enhanced the manuscript, ensuring it is more aligned with the main objectives, as you suggested. Your feedback has played a crucial role in guiding us towards improving the quality of our review. Thank you.

Comments on the Quality of English Language

The manuscript requires significant improvements in language use, from the abstract to the conclusion. The current state of the text makes it difficult to understand the main ideas clearly. A thorough grammatical revision by a native English speaker or a professional language editing service is strongly advised.

Response:

Thank you for your input. We understand how crucial clear language is in academic writing. The first and second authors are proficient in English for academic purposes, and the corresponding author has extensive experience as an associate professor in a US university. We've gone through the manuscript meticulously and made significant improvements to ensure the language is clear and the ideas flow smoothly.

Thank you once again for your time and consideration.

Sincerely yours,

Abdul Wakeel

Reviewer 2 Report

Comments and Suggestions for Authors

The article "Reviving Natural Rubber Synthesis via Native/Large Nano-discs" discussed an interesting topic and was well written. Below, commendations should be considered: 

1. The authors proposed a new solution for natural rubber manufacturers (RTase complex). On what basis was this proposal based?

2. I noticed that the authors used the pronoun "we" many times; I prefer to avoid such a writing form.

3. Please provide a diagram describing your proposal.

4. The reference style should follow the MDPI style.

5. Since you have a new proposal, it is better to compare your method with other methods on the same topic. 

6. Please add references to the replotted figures, for example, Figure 3.

Comments on the Quality of English Language

Good

Author Response

Response to Reviewer 2

We sincerely appreciate your thorough review and insightful feedback throughout this process. Your valuable input has undoubtedly strengthened our manuscript, and we are grateful for the opportunity to improve our work with your guidance. Here are the responses to each of your comments one by one:

Comments and Suggestions for Authors

The article "Reviving Natural Rubber Synthesis via Native/Large Nano-discs" discussed an interesting topic and was well written. Below, commendations should be considered: 

  1. The authors proposed a new solution for natural rubber manufacturers (RTase complex). On what basis was this proposal based?

Response:

Thank you for your feedback. The proposal for utilizing "natural nanodiscs" and large nano-discs for isolating and reconstituting the rubber transferase (RTase) complex is based on both experimental experience and insights from existing literature.

Firstly, the use of natural nanodiscs as an alternative to detergents for isolating the RTase complex draws from the understanding that detergents can disrupt the native structure and function of membrane proteins. Natural nanodiscs, composed of native membrane lipids, offer a more physiologically relevant environment for maintaining the integrity and activity of membrane-associated proteins like the RTase complex.

Secondly, the adaptation of large nano-discs for incorporating and reconstituting the RTase complex builds upon the concept of nanotechnology in biochemical research. Large nano-discs provide a stable platform for studying membrane proteins and can accommodate larger protein complexes like the RTase complex, potentially preserving its native conformation and function during reconstitution experiments.

Moreover, the proposal is informed by the recognition of the longstanding challenges in isolating and characterizing the RTase complex despite numerous attempts. By synthesizing insights from past research endeavors and applying innovative methodologies, such as the use of natural nanodiscs and large nano-discs, we aim to offer a promising solution to enhance our understanding of the RTase complex and its role in natural rubber synthesis

  1. I noticed that the authors used the pronoun "we" many times; I prefer to avoid such a writing form.

Response:

In the revised manuscript, we have made a concerted effort to minimize the unnecessary use of the pronoun "we." We believe this adjustment enhances clarity and readability while still effectively communicating our findings and arguments. We appreciate your attention to detail and hope that you find the revisions satisfactory.

  1. Please provide a diagram describing your proposal.

Response:

We appreciate your attention to detail and understand the importance of visual aids in clarifying our proposal. We apologize for any confusion caused by the previous version of our manuscript. We'd like to inform you that we have addressed this concern in the revised manuscript. We have elaborated on the structural models of the proposed idea in Figures 3-5, providing more detailed legends to enhance clarity. We believe these revisions effectively address your suggestion and hope that they contribute to a better understanding of our work.

  1. The reference style should follow the MDPI style.

Response:

The reference style formatting has been revised as per MDPI style.

  1. Since you have a new proposal, it is better to compare your method with other methods on the same topic. 

Response:

Thank you for your helpful feedback. We've revised the manuscript thoroughly to tackle your concerns. Based on your suggestion to compare our method with others in the field, we've included more literature and rephrased sections to explain each technique clearly and why we're proposing our new technology for RTase isolation. Specifically, we've focused on partially productive techniques like using detergent-washed rubber particles and employing nanodiscs in RTase complex isolation. Additionally, we've given a brief overview of alternative approaches in the last two decades and the last paragraphs of the introduction. We've also made sure to explain why we chose our technique before discussing the methodologies. This helps readers understand why our approach is advantageous for RTase isolation. We believe these improvements address your concerns and give a clearer view of the novelty and importance of our proposed methodology. Thank you once again for your thoughtful review.

  1. Please add references to the replotted figures, for example, Figure 3.

Response:

We've added references to the replotted figures, including Figure 3, in the revised manuscript. Additionally, we've expanded the legends significantly to enhance clarity and understanding.

Thank you once again for your time and consideration.

Sincerely yours,

Abdul Wakeel

Reviewer 3 Report

Comments and Suggestions for Authors

The manuscript entitled ‘’ Reviving Natural Rubber Synthesis via Native/Large Nanodiscs’’ considers the issue of natural rubber synthesis. The approach involves the detergent-free isolation of the RTase complex with the native membrane lipids followed by its reconstitution on liposomes. There is a definite novelty. Anyway, some questions arise after reading:

1.     It seems that the authors did not use the Polymers template, but simply placed the text designed for another journal in the Polymers’ file 2021, which was downloaded some time ago. Required sections are missing. The design of the pictures, the list of references, the font: everything indicates this. Such a careless attitude is surprising.

 Line: 47, 52, 305-314, 393-397, please see.

The Polymers template 2024: In the text, reference numbers should be placed in square brackets [ ] and placed before the punctuation; for example [1], [1–3] or [1,3]. For embedded citations in the text with pagination, use both parentheses and brackets to indicate the reference number and page numbers; for example [5] (p. 10), or [6] (pp. 101–105).

2.     The reference list in reviews usually contains more than 100 references.

Perhaps the submitted manuscript is more suitable for Biochemistry or Biomacromolecules journals, but this is at the discretion of the Academic Editor(s).

Comments on the Quality of English Language

Minor editing.

Author Response

Response to Reviewer 3

We sincerely appreciate your thorough review and insightful feedback throughout this process. Your valuable input has undoubtedly strengthened our manuscript, and we are grateful for the opportunity to improve our work with your guidance. Here are the responses to each of your comments one by one:

Comments and Suggestions for Authors

The manuscript entitled ‘’ Reviving Natural Rubber Synthesis via Native/Large Nanodiscs’’ considers the issue of natural rubber synthesis. The approach involves the detergent-free isolation of the RTase complex with the native membrane lipids followed by its reconstitution on liposomes. There is a definite novelty. Anyway, some questions arise after reading:

  1. It seems that the authors did not use the Polymers template, but simply placed the text designed for another journalin the Polymers’ file 2021, which was downloaded some time ago. Required sections are missing. The design of the pictures, the list of references, the font: everything indicates this. Such a careless attitude is surprising.

 Line: 47, 52, 305-314, 393-397, please see.

The Polymers template 2024: In the text, reference numbers should be placed in square brackets [ ] and placed before the punctuation; for example [1], [1–3] or [1,3]. For embedded citations in the text with pagination, use both parentheses and brackets to indicate the reference number and page numbers; for example [5] (p. 10), or [6] (pp. 101–105).

Response:

We've thoroughly addressed the concerns raised. All required sections have been included, and we've adhered to the Polymers template guidelines, ensuring proper formatting, citation style, and overall presentation. Thank you for bringing this to our attention!

  1. The reference list in reviews usually contains more than 100 references.

Response:

We appreciate your consideration regarding the reference list in our review. We acknowledge that while it's typical for such lists to contain over 100 references, our focus has been on ensuring the inclusion of pertinent and impactful literature relevant to our topic. As a result, we have curated a reference list comprising 96 key sources that we believe offer comprehensive coverage of the subject matter at hand. We are confident that these selected references provide a robust foundation for our review and contribute significantly to the scholarly discourse on the topic.

Perhaps the submitted manuscript is more suitable for Biochemistry or Biomacromolecules journals, but this is at the discretion of the Academic Editor(s).

Response:

Thank you for your input. Our manuscript, "Reviving Natural Rubber Synthesis via Native/Large Nanodiscs," explores the complexities of natural rubber production, which ties in well with the focus of the special issue, "Advanced Natural Polymers: Synthesis, Characterization, and Applications." We are confident that our research provides valuable contributions to this area and would be a suitable inclusion in the journal's compilation.

Thank you once again for your time and consideration.

Sincerely yours,

Abdul Wakeel

Round 2

Reviewer 1 Report

Comments and Suggestions for Authors

After thoroughly reviewing each response provided by the authors of the manuscript and the revised version of the manuscript, I consider that the article does not fulfill the stated objective of the manuscript as expressed by the authors themselves. Once again, the article is difficult to comprehend regarding how information is presented. It is challenging to discern the contribution of this article as a review.

In fact, with the information provided by the authors (see page 3, line 155 to page 4, line 188), it is necessary to clarify the contribution of this review compared to the previous publication (Umar, 2023). Once again, there is a lack of comprehensive analysis of existing scientific reports. The article fails to identify key gaps and ongoing challenges within the relevant field. The conclusion remains overly general, and, with the conclusion, the document becomes even less comprehensive, as the manuscript does not deliver what is summarized in it.

Additionally, some references are not in the same format; they lack homogeneity. Some references are numbered, while others are in a different format.

Comments on the Quality of English Language

Review details and format.

Reviewer 3 Report

Comments and Suggestions for Authors

Accept in present form